# An encrusting kleptoparasite-host interaction from the early Cambrian

Zhifei Zhang [1✉], Luke C. Strotz [1✉], Timothy P. Topper [1,2✉], Feiyang Chen[1,3], Yanlong Chen[1], Yue Liang[1], Zhiliang Zhang[1,3], Christian B. Skovsted[1,2] & Glenn A. Brock [1,3]

Parasite–host systems are pervasive in nature but are extremely difficult to convincingly identify in the fossil record. Here we report quantitative evidence of parasitism in the form of a unique, enduring life association between tube-dwelling organisms encrusted to densely clustered shells of a monospecific organophosphatic brachiopod assemblage from the lower Cambrian (Stage 4) of South China. Brachiopods with encrusting tubes have decreased biomass (indicating reduced fitness) compared to individuals without tubes. The encrusting tubes orient tightly in vectors matching the laminar feeding currents of the host, suggesting kleptoparasitism. With no convincing parasite–host interactions known from the Ediacaran, this widespread sessile association reveals intimate parasite–host animal systems arose in early Cambrian benthic communities and their emergence may have played a key role in driving the evolutionary and ecological innovations associated with the Cambrian radiation.

[1] State Key Laboratory of Continental Dynamics, Shaanxi Key Laboratory of Early Life & Environments and Department of Geology, Northwest University, 710069 Xi'an, China. [2] Department of Palaeobiology, Swedish Museum of Natural History, SE-10405 Stockholm, Sweden. [3] Department of Biological Sciences, Macquarie University, Sydney, NSW 2109, Australia. ✉email: elizf@nwu.edu.cn; lukestrotz@nwu.edu.cn; timothy.topper@nwu.edu.cn

Parasitism is an enduring symbiotic relationship in which the parasite is nutritionally dependent upon the host for at least part of its life cycle, increasing its own fitness in the process and directly impinging upon the biological fitness of the host[1–4]. Parasite–host interactions form a significant proportion of the biotic interactions in extant global ecosystems, influencing many characteristics of species networks including behavior, population structure, and ecological function[5–9]. The antagonistic relationship between parasites and hosts has also been proposed as the primary mechanism leading to the evolution and maintenance of sexual reproduction, due to the negative frequency-dependent selection associated with parasitism[10]. Despite its obvious importance, the origins and early evolution of metazoan parasitism remains enigmatic[4,11,12]. Molecular phylogenies predict the emergence of parasitic clades in the Cambrian[13,14] and putative instances of shell damage, shell scarring and occasional bioclaustration from the early Cambrian[15,16] represent circumstantial evidence that hint at possible parasitism, but the rarity of well-preserved specimens precludes decisive identification of parasite–host interactions in the earliest Phanerozoic. Possible examples of epibiontism[17,18], commensal infestation[19] and hitchhiking[20] are also known from the early Cambrian, but none of these constitute definitive instances of parasitism with a clear

negative biological effect on the host. This absence of clear evidence for parasitism in the earliest animal communities may, in part, be due to a lack of cross-sectional quantitative analyses on Cambrian material of the type that have been demonstrated as necessary to identify and discriminate instances of animal parasitism in deep time[21,22].

The early Cambrian (Stage 4) Guanshan Konservat-Lagerstätte occurs mostly in the lower 40 m of the Wulongqing Formation, which crops out over a geographically wide area in eastern Yunnan, located in southern China (Supplementary Fig. 1). The Guanshan Biota is unusual in being proportionately dominated by brachiopods[23], and so, is strongly differentiated from other Cambrian Konservat-Lagerätten such as the Chengjiang, Sirius Passet, Emu Bay Shale and the Burgess Shale, which are euarthropod-dominated assemblages[24–27]. The organophosphatic linguliform brachiopod *Neobolus wulongqingensis* sp. nov. is the most numerically abundant taxon in the Wulongqing Formation, with many thousands of specimens forming dense concentrations of monotypic, mostly conjoined shells, clustered closely on bedding plane surfaces (Fig. 1a and Supplementary Figs. 2–5). Remarkably, many of the brachiopod shells are encrusted with elongate, tapering biomineralized tubes (Fig. 1 and Supplementary Figs. 2 and 3). Symbiotic relationships such as this are

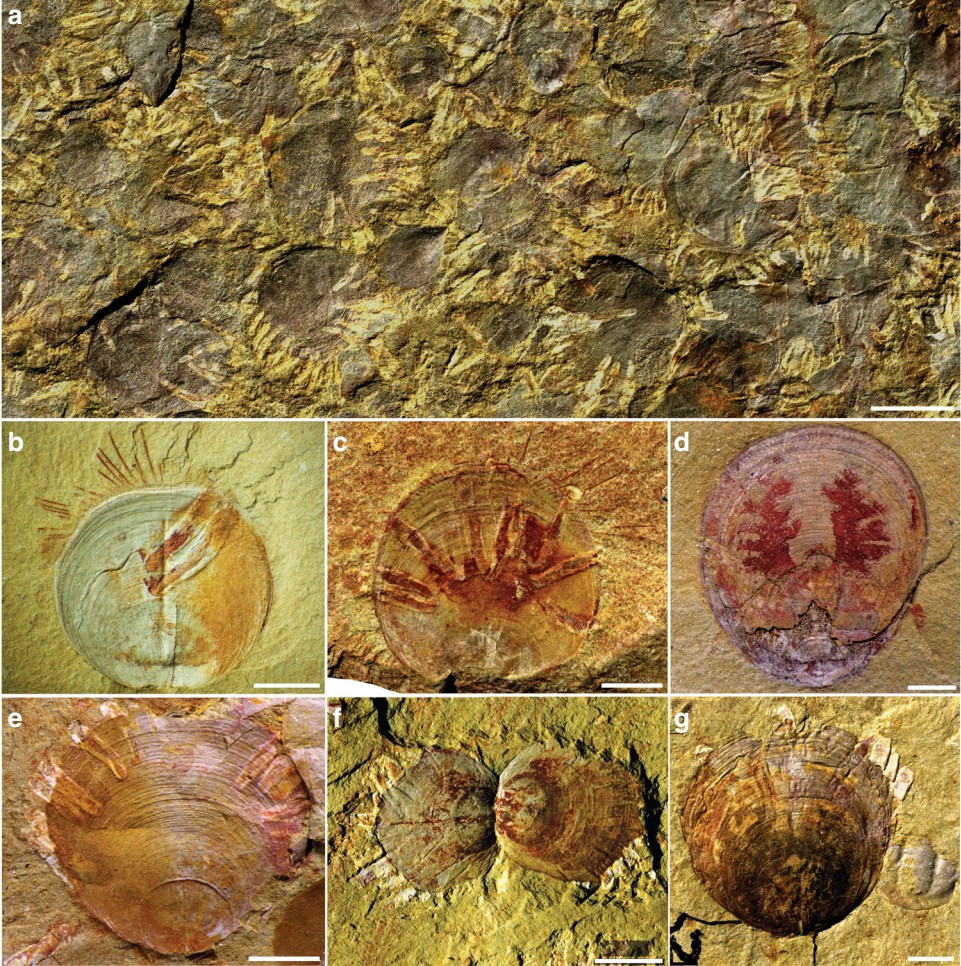

**Fig. 1 The brachiopod *Neobolus wulongqingensis* sp. nov., with associated obligate, encrusting kleptoparasitic tubes. a** ELI GB-N-0301, densely aggregated valves of *N. wulongqingensis* forming distinctive shell beds with their associated kleptoparasites. Scale bar 4 mm. **b**, **c** Specimens of *N. wulongqingensis* with varying numbers of encrusting kleptoparastic tubes (see Supplementary Note 1 for details); **b** ELI GB-N-0650, **c** ELI GB-N-0648-5. **d** ELI GB-N-0008, *N. wulongqingensis* with baculate mantle canals preserved. **e** ELI GB-N-0261-18, *N. wulongqingensis* with encrusting kleptoparastic tubes. **f** ELI GB-N-0255-6, internal view of a pair of conjoined valves with kleptoparasitic tubes encrusted to both valves **g** ELI GB-N-0869-2-1. *N. wulongqingensis* with encrusting kleptoparasitic tubes and trilobite cranidium (exuviae) lacking attached tubes. Scale bars 2 mm, unless otherwise stated.

seldom directly observed in the fossil record because taphonomic biases generally impede the preservation of direct interaction between organisms (see refs. [4,21,22,28,29] for exceptionally preserved cases in younger strata). The high-fidelity preservation and great abundance of specimens in the Wulongqing Formation (Fig. 1 and Supplementary Figs. 2, 3, 5) provides a rare opportunity to investigate this unique interaction between a brachiopod host and their associated encrusting tube-dwelling organisms.

Here we assess differences in biomass between brachiopod individuals of the species *N. wulongqingensis* encrusted with tubes and those individuals lacking tubes, with biomass representing a proxy for the biological fitness of an individual. Our analyses suggest that the tube-dwelling organisms reduced the biological fitness of the host and, when considered in combination with observations of the preferred growth orientation of the encrusting tubes, these results suggest the interaction between the tube-dwelling organisms and their host brachiopod represents kleptoparasitism. This instance in a Cambrian epibenthic marine community likely represents the oldest known parasite–host relationship in the fossil record and reveals that parasite–host interactions emerged in conjunction with the rise of the earliest animal communities during the Cambrian radiation.

## Results and discussion

**A symbiotic interaction.** The preservation of marginal chaetae (Fig. 1b, c and Supplementary Figs. 3a–c and 4a, c, e), mantle canals (Fig. 1d), visceral areas (Supplementary Figs. 2b and 4f) and, rarely, the lophophore (Supplementary Fig. 4f–h; see also ref. [23]) in the brachiopods indicates rapid burial and minimal transport by episodic obrution deposits[18]. Despite this, the soft body of the tube-dwelling animal is not well-preserved, and its biological affinities are not self-evident. The greyish-white tubes, normally flattened by post depositional compaction, are immediately apparent (Fig. 1a–c, e–g and Supplementary Figs. 2 and 3). The tubes, some with preserved accretionary growth increments

(Supplementary Fig. 3e–f), encrust the exterior of both dorsal and ventral valves of *N. wulongqingensis* (Fig. 1f and Supplementary Fig. 3d) with the open apertures exclusively oriented toward the anterior commissure of host brachiopods, indicating an intimate, life-long, in-vivo association. The tubes exclusively encrust the exterior of the host shell, which occasionally shows signs of minor damage or disruption of shell growth lines, but there is no evidence of boring into the interior of the brachiopod by the tube-dwelling organism. The tubes are not found attached to any other hosts or substrates, such as the trilobite (Fig. 1g) or palaeoscolecid exuviae that occasionally occur in the shell beds. Consequently, we interpret this interaction as representing an obligate relationship, as defined by Poisot et al.[30], as there is no evidence to suggest that the tube-dwelling organisms can adopt a free-living lifestyle in the absence of their brachiopod host.

**Impact on host biomass.** Bayesian estimation analysis[31] demonstrates that a credible difference in biomass exists between brachiopods with encrusting tubes (n = 205) compared to those without (n = 224). There is no overlap in the 95% highest density interval (HDI) of the posterior distribution for the means of the two groups and the HDI for effect size does not overlap with zero (Fig. 2a, b). Mean biomass for individuals with encrusting tubes is thus credibly lower than for those without tubes. A null hypothesis significance testing approach also identifies a significant difference between encrusted and non-encrusted individuals with a small effect size (Wilcoxon–Mann–Whitney $W = 18,186$, $P = 0.0002$, Cliff's Delta = −0.208). We therefore contend that individual brachiopods encrusted with tubes have reduced fitness when compared with their non-encrusted counterparts. On the basis of the difference in the values for mean biomass between the two groupings, encrustation results in a 26.08% reduction in overall fitness across the entire measured cohort.

Although our analyses indicate that brachiopods with encrusting tubes are reduced in biomass compared to those without,

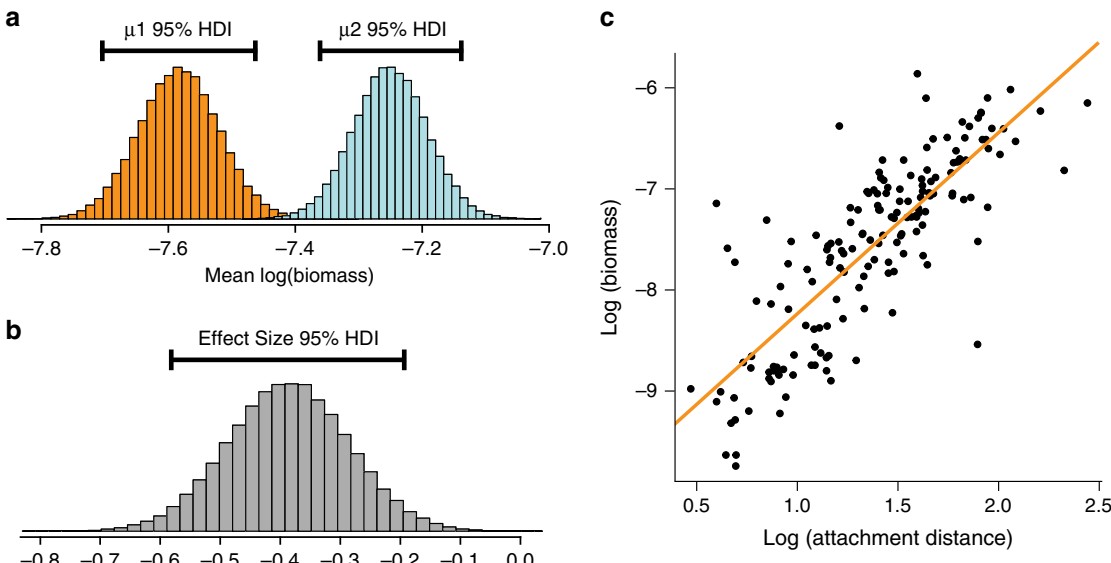

**Fig. 2 Results of analyses demonstrating encrusted tubes were parasitic. a** Posterior distribution of mean biomass derived from Bayesian estimation for brachiopods without attached tubes (μ1; left) versus those brachiopods with encrusted tubes (μ2; right). HDI denotes highest density interval and represents credible values for mean biomass for each grouping. **b** Posterior distribution of effect size for μ1 versus μ2 derived from Bayesian estimation. HDI exceeds 0, indicating that a credible difference exists between the mean values for brachiopods with encrusted tubes versus those brachiopods without tubes. **c** Plot of Attachment Distance versus Biomass. Attachment distance from the posterior margin of *N. wulongqingensis* represents a proxy for the duration of the symbiotic relationship between an individual brachiopod and its associated encrusted tubes. Correlation between these two variables therefore indicates that those brachiopods with enduring symbiotic relationships are reduced in biomass in comparison to those where time of attachment has been short. Source data provided in Supplementary Data 1 and 2.

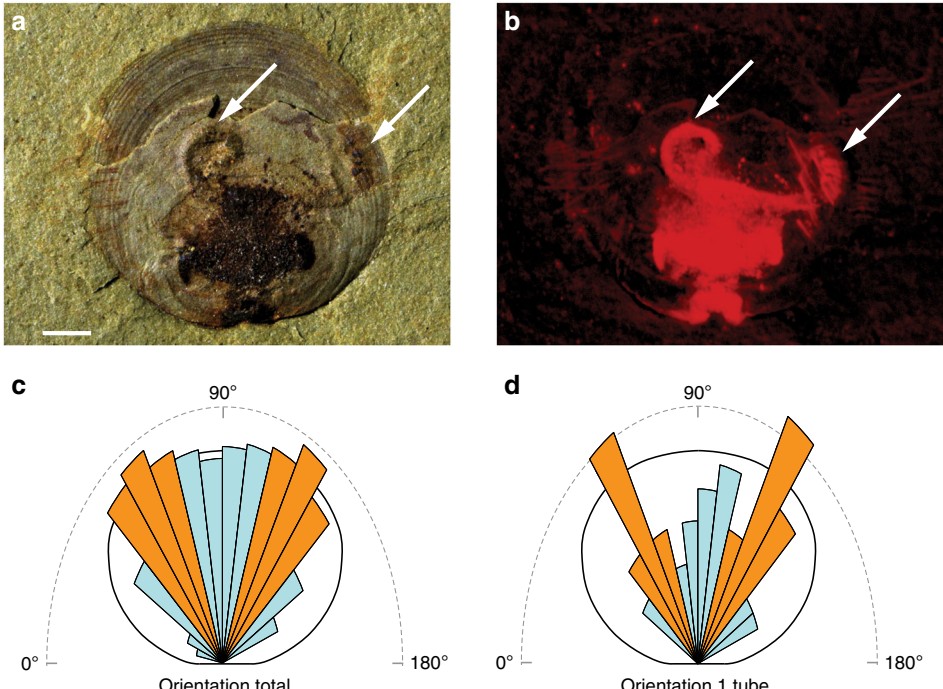

**Fig. 3 Evidence demonstrating the associated encrusted tube-dwelling organisms were kleptoparasitic. a** Shell interior of specimen ELI GB-N-0595A of *N. wulongqingensis* from Wuding (Supplementary Fig. 1a, Locality 2) showing the presence of a paired spirolophe lophophore (as indicated by white arrows). Scale bar is 1 mm. **b** Micro-XRF elemental mapping of Fe for ELI GB-N-SJJ-0595A provides a high contrast image of the spirolophe lophophore (as indicated by white arrows). **c**, **d** Rose diagrams of attached tube orientation for: **c** All measured individuals of *N. wulongqingensis* (*n* = 146) and; **d** *N. wulongqingensis* individuals with only one attached tube (*n* = 31). Each division represents a 10° interval. Intervals colored in orange are those that correspond to the inhalant laminar currents generated by *N. wulongqingensis*. For all numbers of attached tubes, orientations that align with inhalant laminar currents are preferred but for individuals with only one attached tube, where the symbiont has all available orientations still available, orientations that align with the inhalant laminar currents are strongly preferred.

there is no clear relationship between the biomass of host individuals and increasing numbers of encrusted tubes per individual (Supplementary Fig. 6a). In some symbiotic relationships, the impact on the host is amplified depending on the number of parasites present[8], but this relationship can be highly variable[32]. For our dataset, the biomass of the brachiopod host decreases when a single tube is encrusted to the shell surface, but no further decline is associated with an increasing number of tubes (Supplementary Fig. 6a). Both proxies for increasing total parasite load also show no correlation with biomass (adjusted $R^2$ for both = −0.0003; Supplementary Fig. 6b, c). This suggests the tube-dwelling organism did not directly inhibit the feeding capability of the host, as larger numbers of parasites do not result in decreased fitness. However, a significant relationship exists between the attachment point of the encrusting tubes and the biomass of the brachiopod host ($P$ = 2.2e−16, $F$ = 327.6, 165 degrees of freedom, $R^2$ = 0.66; Fig. 2c), indicating encrustation earlier in ontogeny results in greater reduced biomass relative to hosts that have been infected at later ontogenetic stages, regardless of the number of symbionts present (Supplementary Fig. 6). In living brachiopods, smaller individuals generally display an increased growth rate compared to larger individuals[33]. It would therefore be expected that the impact on fitness would be greater for host individuals that are settled by parasites during earlier ontogenetic stages. The increase in median attachment distance for larger numbers of symbionts (Supplementary Fig. 6d) and the larger size of specimens with greater than four tubes (Supplementary Fig. 6a) establishes that higher infection rates can only occur when brachiopod hosts have already managed to grow to larger adult sizes and there is sufficient brachiopod shell surface area to accommodate a larger number of encrusting tubes.

This also indicates that the tube-dwelling organisms do not preferentially encrust smaller brachiopod individuals, as brachiopods are clearly encrusted in large numbers later in their ontogeny, when they have reached larger sizes.

Our analyses demonstrate that the tube-dwelling organism directly impinges upon the biological fitness of the host, supporting the assertion that the encrusting tube-dwelling organisms are parasitic, rather than being either mutualistic or commensal with the brachiopod host. A reduction in host biomass or growth rate has been directly attributed to the presence of a parasite in a variety of extant symbiotic relationships[6,8,9]. Parasites typically increase the energetic requirements of infected organisms, as the host must generate sufficient energy to not only maintain its own requirements but also the needs of the parasite[6]. This commonly leads to hosts with decreased biomass when compared with uninfected individuals. This result represents the first definitive and statistically supported instance of parasitism from the Cambrian and indicates that parasite–host systems were well established by Cambrian Stage 4, suggesting this type of interaction probably emerged even earlier during the main pulse of the Cambrian radiation.

Variations in biomass between individuals and assemblages of the same species have also been previously attributed to regional variation in environmental stressors[6]. All specimens of *N. wulongqingensis* included in this analysis occur in dense aggregations (estimated 60,000 individuals per m²—Supplementary Fig. 5) from the same geographic locality (Supplementary Fig. 1a, locality 3) and stratigraphic package with similar sedimentological features[23] subject to similar environmental and depositional conditions. Consequently, the reduced biomass

of tube-encrusted *N. wulongqingensis* individuals cannot be attributed to environmental factors and a parasitic affect is the most strongly supported probable cause.

**A kleptoparasitic relationship.** In all instances, the apertures of tubes are orientated toward the brachiopod commissure, spanning an arc (plan view) of ~150° (Fig. 3). No tubes have been observed orientated toward the hinge line of the brachiopod. Tubes consistently grow beyond the commissural margin of *N. wulongqingensis* into, and slightly above but rarely beyond, the brachiopod chaetal fringe (Fig. 1b, c and Supplementary Fig. 3a–c). Critically, the dominant growth direction of the tubes aligns tightly along a vector between 40° and 70° either side of the median plane of symmetry of the brachiopod (Fig. 3c; Supplementary Data 3); this alignment is most pronounced in shells with a single encrusting tube (Fig. 3d) but the same orientation pattern occurs in shells with all numbers of tubes (Supplementary Fig. 7), strongly supporting a preferential growth direction in the tubes toward the antero-lateral margin of the brachiopod shell.

Five specimens of *N. wulongqingensis* from Wuding Quarry (Supplementary Fig. 1a, Locality 2) preserve a partial spirolophe lophophore (Fig. 3a, b and Supplementary Fig. 4f–h). A spirolophe lophophore produces two separate inhalant laminar feeding currents at the antero-lateral edge of the shell margin[34] that match the preferred orientation and growth position of the encrusting tubes on shells of *N. wulongqingensis* (Fig. 3c, d and Supplementary Fig. 7). The preferred orientation of growth demonstrates that the tube-dwelling organisms were not purely utilizing the brachiopod as a hard substrate on which to construct their tubes. This data when combined with the demonstrated empirical cost to the host in the form of reduced biomass (Fig. 2a, b), strongly supports kleptoparasitic behavior[35]. Kleptoparasitism is a form of competition, where food that is either already in the possession of the host or which the host has expended energy on obtaining and capture is imminent, is stolen by the parasite[35]. In our scenario, this involves the tube-dwelling organisms acting as intercept feeders—stealing a portion of the brachiopod feeding stream before it reached the chaetal fringe (Fig. 4). Iyengar[35] recognized six distinct morphological, behavioral and physiological criteria that characterize living sedentary/sessile kleptoparasitic interactions. At least five of these criteria can be directly applied to the relationship between the encrusting tube-dwelling organism and *N. wulongqingensis* (Table 1) further reinforcing a kleptoparasitic relationship.

Kleptoparasitism is rarely identified in the fossil record[22], and no instances of kleptoparasitism, as far as we are aware, have been proposed for Cambrian communities[11,12]. Detailed empirical investigations of the energetic and nutritional cost of kleptoparasitism to the host, even for extant systems, are few[8,35]. For this reason, it is currently difficult to assess if the reduction in host fitness (~26%) we detect for *N. wulongqingensis* is typical of sessile kleptoparasitic relationships. Brachiopods are particularly vulnerable to exploitation by kleptoparasites, since active filter feeding represents the greatest energy expenditure in the life of brachiopods[36], and the time lag between collection and ingestion of nutritionally beneficial particles also provides potential for other organisms to exploit this resource[35]. Combined with the fact that the biotic interaction we document is interpreted as obligate for the parasite, this suggests that the effect we observe is likely greater than would be the case in facultative kleptoparasitic associations. Intriguingly, obligate kleptoparasitism is exceedingly rare in modern marine systems[35], which might suggest that this novel ecological relationship is always rare in benthic communities or has been secondarily lost some time during the Phanerozoic.

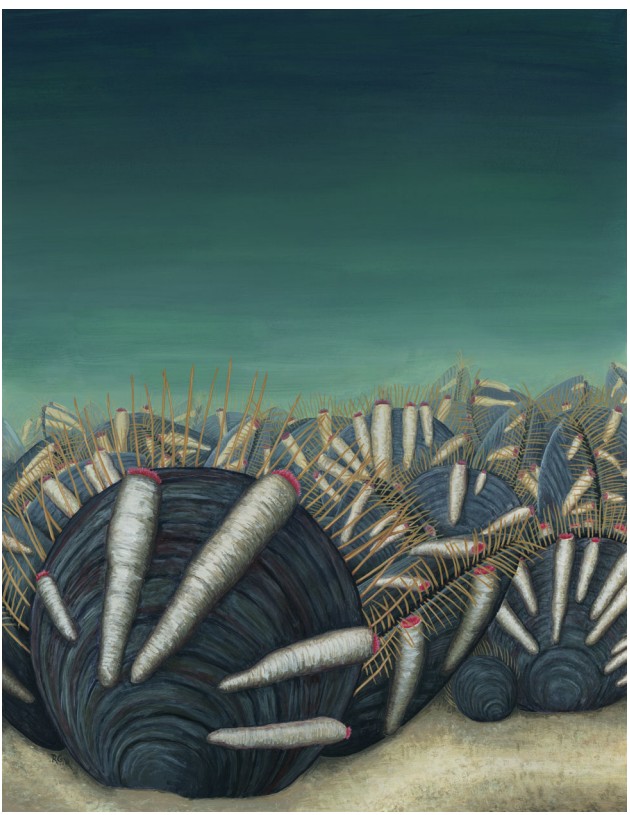

**Fig. 4 Artist's reconstruction of the Wulongqing Formation benthic community.** Reconstruction shows the dense aggregations of monotypic *Neobolus wulongqingensis* forming benthic 'meadows' on the soft sediment with their associated obligate encrusting kleptoparasitic tube-dwelling organisms (Artist: Rebecca Gelernter of Near Bird Studios).

Verification of this kleptoparasitic relationship reveals that the heritage of parasite–host interactions can be traced back more than half a billion years to the rise of bilaterian animal communities during the Cambrian and further establishes the importance of the early Cambrian as a primary source of ecological novelty. Antagonistic biotic interactions have also been proposed as the drivers of widespread evolutionary phenomena such as the maintenance of sexual reproduction and genetic polymorphism at disease loci[10,37]. Both of these are known drivers of biodiversity increase, suggesting that the already established presence of parasitic relationships in Cambrian communities potentially had a fundamental role in the upsurge in evolutionary innovation associated with the Cambrian Radiation.

**Systematic paleontology.** Order Lingulida Waagen, 1885
Family Neobolidae Walcott and Schuchert in Walcott, 1908
Genus *Neobolus* Waagen, 1885
*Neobolus wulongqingensis* sp. nov. Zhang, Strotz, Topper, and Brock
Etymology: After the Wulongqing Formation where the fossils are recovered.
Holotype: ELI B-GSN-0377 (Supplementary Fig. 4a) from the Gaoloufang section.
Other material: ELI B-GSN-0001-0625.
Stratigraphy and distribution: Wulongqing Formation (*Paleolenus* trilobite zone), Cambrian Stage 4. Specimens discovered in wide area of eastern Yunnan, comprising the Gaoloufang section in Kunming, Kanfuqing section in Malong, Shijiangjun sections in Wuding, and Dahai section in Huize (Supplementary Fig. 1).

| Table 1 Characteristics and requirements of extant sedentary/sessile kleptoparasitic interactions (from ref. [35]; left) compared with early Cambrian brachiopod–tube interaction from the lower Cambrian (Stage 4) Guanshan Konservat-Lagerstätte (right). | |
| --- | --- |
| **Extant sedentary kleptoparasitic interaction** | **Guanshan brachiopod-encrusted tube interaction** |
| Interactions are long in duration, the kleptoparasite utilizes few hosts within its lifetime (sometimes only one) | Obligate and enduring interaction between kleptoparasitic encrusting tube-dwelling organisms and adult organophosphatic brachiopod host |
| Extreme negative impact on the host is not possible because of limited ability to find a new host if the present one dies | Measurable negative non-fatal impact on host (Fig. 2); encrusting life habitus limits association to single host over lifetime |
| One host might be sufficient | Obligate parasite–host interaction |
| Individual items do not need to be large, as not much energy is expended by kleptoparasite to steal food and host feeding is continuous. Regular access to adequate food is necessary | Kleptoparasite an intercept filter feeder; steals proportion of inhalant particulate food stream generated by brachiopod. Brachiopod feeding almost continuous |
| Life span of host exceeds a sessile kleptoparasite's time to sexual maturation or hosts aggregate in heterochronous assemblages if kleptoparasite is sedentary | Life span of brachiopod and tube unknown, death of host kills parasite; host forms dense concentrations of clustered shells of variable size. Host and kleptoparasitic tube are sessile |
| Host must not consume a large proportion of the kleptoparasite's offspring, either during larval emergence or settlement | Unknown. Enduring geographically widespread occurrence of unique, obligate brachiopod–tube shell beds suggests successful larval recruitment for both taxa |

Diagnosis: Adult shell subcircular, no visible pits or pustules on surface, peripherally ornamented with distinct growth lines; metamorphic shell, average of 2396 in width and 1907 μm in length ($n = 13$); ventral pseudointerarea orthocline to apsacline with wide and triangular pedicle groove; ventral propareas vestigial or indistinguishable; dorsal pseudointerarea forming narrow, crescent-shaped rim; ventral visceral field short, slightly thickened and not extending beyond midvalve; dorsal interior with long median septum extending to or beyond 1/3 valve length. Short spirolophe lophophore present (Fig. 3a, b and Supplementary Fig. 4f–h). Marginal chaetae long and closely packed, extending up to 4.6 mm beyond the valve margin, forming a distinctive chaetal fringe (Supplementary Fig. 4a, c–e); Ventral mantle canals baculate (Fig. 1d).

## Methods

**Studied material.** High-density shell-bed concentrations of *N. wulongqingensis* and associated encrusted tube-dwelling organisms from the Guanshan Konservat-Lagerstätte were sampled as slabs of fine grained mudrocks and siltstones from thin-bedded mudrock layers within the basal 20 m of the Wulongqing Formation (formerly Wulongqing Member of Canglangpu Formation) from the Gaoloufang section, near Kunming (Supplementary Fig. 1a). Previous intensive excavations of the Guanshan Konservat-Lagerstätte from the Gaoloufang and Kanfuqing sections near Kunming and Malong[23] have revealed a biota dominated by a rich assemblage of trilobites, bivalved arthropods, palaeoscolecid worms and an assortment of non-mineralized soft-bodied organisms. Aggregated high-density shell-bed concentrations of *N. wulongqingensis* have now been recovered at a number of localities (Supplementary Fig. 1a) but all specimens included in this investigation come from rock slabs collected from the Gaoloufang section (Supplementary Fig. 1a) and deposited in the collections of the Shaanxi Key Laboratory of Early Life & Environments and Department of Geology, Northwest University (Supplementary Data 1 and 2). Of the 429 specimens included in this analysis, 224 specimens were uninfected and 205 specimens were infected by the encrusting tube-dwelling parasites (Supplementary Data 1).

**Measurements.** Specimens were photographed using a Zeiss Smart Zoom 5 Stereomicrographic system and micro-XRF elemental mapping was undertaken using a Bruker M4 Tornado Micro-XRF spectrometer. The maximum linear dimension for each specimen was determined by taking measurements using ImageJ 1.49v[38]. Previous studies have shown that maximum linear dimension is directly correlated with soft tissue mass[39], allowing direct conversion between linear measurements of shell size and biomass. We use measurements of biomass for individual specimens as a proxy for the relative fitness of brachiopod individuals and to identify differences in fitness between individuals with encrusted tubes and those without. We intended to combine these measurements with growth rate to obtain a holistic appraisal of the potential effects of encrusted tubes on brachiopod fitness, but the quality of preservation of *N. wulongqingensis* from the Guanshan Konservat-Lagerstätte is insufficient to consistently discern the very fine individual accretionary growth lines on individual specimens. We therefore confine our analyses to an assessment of potential disparities in biomass. To convert our linear measurements to biomass (specifically ash-free dry weight), we use the scaling coefficient and exponent for brachiopods presented in Payne et al.[40]. Body size measurements and biomass values for all specimens are provided in Supplementary Data 2.

**Statistical analyses.** All analyses of our dataset were undertaken using R 3.6.1[41]. To assess whether a difference in biomass exists between individuals with encrusted tubes compared to those without, Bayesian estimation analysis, using the 'BEST' package[31] generated complete distributions of credible values for group means and effect size (Fig. 2). In addition to this Bayesian approach, we also employ a null hypothesis significance testing approach, in the form of a Mann–Whitney U test, as a secondary assessment of the potential difference between our two groups. Cliff's Delta[42], calculated using the 'effsize' package, is used as a measure of effect size for this secondary analysis.

Measurements of encrusted tube length, width, orientation and distance from margin were all compiled using the approach illustrated in Supplementary Fig. 8. Tube length and width were used to calculate total tube width per individual and the total surface area of each host shell covered by encrusted tubes; both values taken to represent proxies of total potential parasite load. Results of all these measurements are presented in Supplementary Data 2. Potential correlation between brachiopod biomass and total encrusted tube width per individual, total surface area of each host shell covered by encrusted tubes or attachment distance was assessed by fitting a simple linear regression model to the data (Supplementary Fig. 6b, c). Rose diagrams of encrusted tube orientation (Fig. 3 and Supplementary Fig. 7) were generated using the 'circular' package[43].

**Nomenclatural acts.** This published work and the nomenclatural acts it contains have been registered in ZooBank, the proposed online registration system for the International Code of Zoological Nomenclature (ICZN). The ZooBank LSIDs (Life Science Identifiers) can be resolved and the associated information viewed through any standard web browser by appending the LSID to the prefix "http://zoobank.org/". The LSID for this publication is: urn:lsid:zoobank.org:act:387EDCD0-3DA8-4965-8F36-A33032317209.

**Reporting summary.** Further information on research design is available in the Nature Research Reporting Summary linked to this article.

## Data availability

The authors declare that all data supporting the findings of this study are available within the paper and its supplementary information files (specifically Supplementary Data 1 and 2). Specimens are deposited in the collections of the Early Life Institute and Department of Geology, Northwest University (prefix: ELI). Photographic material of the studied material is available from the corresponding authors upon request. The source data underlying Fig. 2 and Supplementary Fig. 6 is provided in Supplementary Data 1–3.

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

## Acknowledgements

Financial support (to NWU group) from the National Natural Science Foundation of China (Grant Numbers 41425008, 41720104002, 41890844 and 41621003), and the Strategic Priority Research Program of the Chinese Academy of Sciences and the 111 project (D17013) is sincerely acknowledged; L.C.S. and T.P.T. are funded by 1000 Talent Program; G.A.B. is funded by a 1000 Talent Shaanxi Province Fellowship. We thank J.-P. Zhai and X.-Y. Ren (in Xi'an) for preparation of the fossils and technical assistance in micro-XRF mappings.

## Author contributions

Z.-F.Z., G.A.B., L.C.S. and T.P.T. designed the study. Z.-F.Z., F.-Y.C., Z.-L.Z., Y.-L.C. and Y.L. completed fieldwork in Kunming and Malong and collected all specimens used in the study. F.-Y.C., Y.-L.C. and Y.L. photographed the specimens and completed EDX and micro-XRF analyses. T.P.T and Y.L. completed all the measurements. L.C.S. and Y.-L.C. completed all statistical analyses. Z.-F.Z., G.A.B., L.C.S. and T.P.T. wrote the manuscript, with input from C.B.S. and all authors. All authors discussed and agreed on interpretation of the data.

## Competing interests

The authors declare no competing interests.
