## [Peer Review File · Nature Communications]

Reviewers' Comments:

Reviewer #1:

Remarks to the Author:

I congratulate you for bringing new, exciting and much needed data on parasitic interactions in the Cambrian. Particularly a quantitative approach you apply in this paper has been greatly missing so far. I would love to see this published, there are just some minor things I would like to see implemented/addressed.

Previous work in the Cambrian: This is a very interesting study which is the first to quantitatively analyze parasitic interactions in the Cambrian, documents a negative effect on their host and document a now probably extinct parasitic interactions which is worth publishing 10times over one of these reasons alone. For some other claims I advise the authors for expressing themselves more carefully – there are other studies who suggest parasitic (and by definition negative effect) on early Cambrian organisms which merit to be mentioned (Bassett, Popov, and Holmer 2004, Peel 2014). It might also be worth highlighting that current phylogenies demonstrate that parasitism is derived in metazoa and that they together with current host associations predict an origin of parasitic metazoa in the Cambrian. What you could however highlight even more is the importance a cross-section analysis brings to interpreting a negative effects and therefore proving a parasitic relationship. Analyses like yours are the way forward to demonstrate a negative in extinct organisms lacking modern analogues (De Baets and Littlewood 2015, Baumiller 2003) and not assume a negative impact if it looks like modern examples.

Prevalence: One can distill the prevalence of infested versus total investigated specimens from the supplementary material, but it would worth discussing this prevalence in the main text and help to back up your interpretation of parasitism. High host specificity and/or high prevalence (between 10 up to 50-70%) are usually taken as a sign of specialist parasites, while very low (<1%) or very high prevalence (100% could signify obligate symbiosis) might signify other causes (Hengsbach 1990, De Baets, Keupp, and Klug 2015). This might also open the window for comparative studies across time and/or space and getting a better handling on collection/sampling biases. When this interaction is missing – is it because too little specimens have been sampled (in cases encrustations are rare) or could differences have rather an evolutionary or ecological reason.

Order of Results/Discussion: Personally, I would prefer to see first the discussion about the in-vivo encrustation and later the relationship with body size as this might make more logical sense. First you need to prove in-vivo encrustations and later the effect on hosts. If encrustations are not in-vivo they might just indicate a

preference of encrusters for particular sizes of substrating. I agree that the most logical explanation is encrustation during life.

Exploration of alternative hypotheses for the size difference: you attribute the size differences almost immediately to differences in fitness but there are potentially other factors (e.g., preservational biases, selective encrustation and/or mortality of juveniles) which would be necessary to be discussed as alternative hypotheses/possibilities in main text even if you can exclude them with the data you have (Huntley and Scarponi 2012). Instead of biomass differences alone in encrusted versus non-encrusted specimens, it would be better to demonstrate an actual impact on growth through a cross-sectional analysis which your data would easily allow (Baumiller and Gahn 2018) and/or perform an actual cost-benefit analysis (Baumiller 2003).

These and additional suggestions can be found in the annotated pdf.

Suggested references:

Bassett, Michael G., Leonid E. Popov, and Lars E. Holmer. 2004. "The Oldest-Known Metazoan Parasite?" *Journal of Paleontology* no. 78 (6):1214-1216.

Baumiller, Tomasz K, and Forest J Gahn. 2018. "The nature of the platyceratid–crinoid association as revealed by cross-sectional data from the Carboniferous of Alabama (USA)." *Swiss Journal of Palaeontology* no. 137 (2):177-187.

Baumiller, Tomasz K. 2003. "Evaluating the interaction between platyceratid gastropods and crinoids: a cost–benefit approach." *Palaeogeography, Palaeoclimatology, Palaeoecology* no. 201 (3–4):199-209. doi: [http://dx.doi.org/10.1016/S0031-0182\(03\)00625-4](http://dx.doi.org/10.1016/S0031-0182(03)00625-4).

De Baets, K., and Littlewood, D. T. J., 2015, *The Importance of Fossils in Understanding the Evolution of Parasites and Their Vectors: Advances in parasitology*, v. 90, p. 1-51.

De Baets, Kenneth, Helmut Keupp, and Christian Klug. 2015. "Parasites of ammonoids." In *Ammonoid Paleobiology: From anatomy to paleoecology. Topics in Geobiology 43*, edited by Christian Klug, Dieter Korn, Kenneth De Baets, Isabelle Kruta and Royal H. Mapes, 837-875. Dordrecht: Springer.

Hengsbach, Rainer. 1990. "Studien zur Paläopathologie der Invertebraten. 1: Die Paläoparasitologie, eine Arbeitsrichtung der Paläobiologie." *Senckenbergiana Lethaea* no. 70 (4-6):439-461.

Huntley, John Warren, and Daniele Scarponi. 2012. "Evolutionary and ecological implications of trematode parasitism of modern and fossil northern Adriatic bivalves." *Paleobiology* no. 38 (1):40-51. doi: 10.1666/10051.1.

Peel, John S. 2014. "Failed predation, commensalism and parasitism on lower Cambrian linguliformean brachiopods." *Alcheringa: An Australasian Journal of Palaeontology*:1-15. doi: 10.1080/03115518.2015.964055.

Reviewer #2:

Remarks to the Author:

This is a very interesting manuscript. The authors described and discussed a unique, life association between a tube-dwelling organism and a new species of brachiopod, *Neobolus wulongqingensis* sp. nov., from the early Cambrian Guanshan Biota, Southwest China. Although symbiotic relationships have been reported several times from exceptionally well-preserved Cambrian fossils, it is generally difficult to determine the exact relationship between the symbionts if it is mutualism, commensalism or parasitism. This study took the advantage of the great abundance of available fossil specimens, cleverly applied statistical methods to explore the interaction between the brachiopod species and the associated tube-dwelling organism, and subsequently concluded this interaction as a type of parasitism (obligate kleptoparasitism). This manuscript is well prepared, the study carried out some detailed measurements and analyses, the descriptions are accurate, and both the discussions and conclusions are generally sound. Therefore, this manuscript presents a welcoming study contributing to our understanding of early symbiotic relationships from the Cambrian, and I would like to support its publication on Nature Communications. However, I have some comments or questions below for further improvements.

Major points:

1) Although the authors convinced me that the interaction between the tube-dwelling organism and *N. wulongqingensis* is likely to be a parasite-host relationship, the conclusion of "obligate kleptoparasitism" might be oversold. From my understanding "obligate parasite" is a parasitic organism that cannot complete its life-cycle without exploiting a suitable host and an "obligate kleptoparasite" derives all of its energy intake from kleptoparasitism. However, we know very little about the nature and life-cycle of the tube-dwelling organisms, so there is no sufficient evidence to support "obligate kleptoparasitism".

2) Although the statistical analysis shows that the tube-dwelling organism likely reduced the fitness of infested brachiopods, it is not clear to me if this reduced fitness was entirely caused by "kleptoparasitism". Are there any other possibilities? For example, did the tube-dwelling organisms bore through the shell of the brachiopods? Could the weight and attachments of these tube-dwelling organisms reduce the movement and feeding efficiency of the brachiopods? If the reduced fitness was entirely caused by being robbed food, why the increasing numbers of tube-dwelling organisms did not cause further decline of the hosts' fitness.

3) It will be nice to introduce and compare with some other examples of interactions between epibiont and host that are observed in the living world today and in other brachiopod fossils. For example, are all these relationships parasitism? Please also see the study by Shiino & Tokuba 2016, Palaeoworld.

4) Some parts of the manuscript lack fluidity, as different sections of the manuscript seem to be slightly disjointed. For example, the Results and Discussion straightly went into the biomass analysis, which is rather abrupt from the previous section. The lack of explanation about the research background and methods can lead to many questions, such as: 1) How many specimens were analyzed? How many of them with tubes? 2) What does biomass mean here? How was it measured from fossil specimens? 3) Why need to measure biomass? Some of these questions were eventually answered in the method section, but I feel that it is still necessary to briefly explain here, or need to cite "(see Material and Methods)".

Minor points:

Page 1, summary: "kleptoparasitism"

This manuscript should attract a wide audience with different background, so it will be helpful to briefly explain some terminologies when they first appear in the manuscript, such as "kleptoparasitism (parasitism by theft)".

Page 2, 2nd paragraph, line 5: "from other Konservat-Lagerstätten"
Change to "from other Cambrian Konservat-Lagerstätten"

Page 2, 3rd paragraph, line 1: "many of the brachiopod"
It will be more accurate to explain the percentage of infested brachiopods

Page 2, 3rd paragraph, line 2: "Whilst the tube-dwelling animal"
Change to "Whilst the soft body of the tube-dwelling animal"

Page 2, 3rd paragraph, line 5: "(Supplementary Fig 3E-F)"
Change to "(Supplementary Fig 3E-F),"

Page 2, 3rd paragraph, line 8: "indicating an intimate, life-long association"
I didn't understand how the statement before indicate a "life-long" association?
Please explain further.

Page 2, 3rd paragraph, line 9: "(Fig. 1g)"
It is better to use arrow or box to mark the trilobite fragment out in the picture

Page 3, 2nd paragraph, line 5: "tube dwelling"

Change to "tube-dwelling", please check throughout the manuscript for consistency

Page 5, 2nd paragraph, line 6: "The preferred orientation...chaetal fringe (Fig. 4)"

This sentence is too long and confusing. Please reword and divide it two sentences.

Page 5, 2nd paragraph, line 10: "strongly supports kleptoparasitic behavior²³"

It is not clear why REF [23] was cited here? Please check.

Page 5, 3rd paragraph, line 10: "and the time lag...exploit this resource"

Please cite REF [24] here

Page 5, 3rd paragraph, line 11: "Combined with the fact the biotic...kleptoparasitic associations."

As mentioned above, I don't fully understand this sentence neither agree with the arguments here. Otherwise, please explain in more detail.

Page 5, last sentence: "Intriguingly, ...during the Phanerozoic"

Same as above, this statement is over pushed

Page 6, last few sentences: "Antagonistic biotic...Cambrian Radiation"

I can understand that the authors want to link their research to a wider implication, but these sentences are not particularly relevant to the study, and the discussion is not rigorous and convincing, more like some throw-in empty comments, so I suggest to delete these sentences; otherwise please reword.

Page 6, Measurements.

The whole paragraph didn't explain what samples have been measured, how they were measured and where is the measurement data

Page 7, Imaging. "Previous studies...Dataset 2"

These sentences should be put into the above Measurements section, and some of them can be used for the Result section as well

Page 7, Statistical analysis. "Measurements of encrusted tube....in Supplementary Dataset 2."

Move to above Measurements section

Best wishes,

Xiaoya Ma

Reviewer #3:

Remarks to the Author:

Dear authors,

I found that the manuscript was well constructed, and answered many of the questions that sprung to mind while reading the manuscript.

My comments and suggestions, referring to more specific and particular areas of the manuscript, are included in annotated file 'Zhang et al_Reviewer_comments', attached to my review.

You may have followed a journal template, but I thought odd the "Results and Discussion" were presented before the "Materials and Methods" section. The supplementary information file is welcome too. All images are clear and sharp; all figures/graphs are of good quality.

Regards

Yves Candela

Reviewer #1

I congratulate you for bringing new, exciting and much needed data on parasitic interactions in the Cambrian. Particularly a quantitative approach you apply in this paper has been greatly missing so far. I would love to see this published, there are just some minor things I would like to see implemented/addressed.

Previous work in the Cambrian: This is a very interesting study which is the first to quantitatively analyze parasitic interactions in the Cambrian, documents a negative effect on their host and document a now probably extinct parasitic interactions which is worth publishing 10times over one of these reasons alone. For some other claims I advise the authors for expressing themselves more carefully – there are other studies who suggest parasitic (and by definition negative effect) on early Cambrian organisms which merit to be mentioned (Bassett, Popov, and Holmer 2004, Peel 2014).

We thank Referee #1 for recommending our manuscript for publication. Referee #1 is correct that there have been previous claims of Cambrian parasite-host relationships but in all cases, they are poorly preserved or rely on circumstantial evidence and so the claims cannot be adequately tested quantitatively. Our paper excluded claims of Cambrian parasitism using a non- **quantitative approach**, but we accept such claims should not be ignored and have followed the advice of Reviewer #1 and have added these in accordingly. We also include citations to previous examples of kleptoparasitism recorded in the fossil record (although not in the Cambrian) and include relevant citations that highlight these instances.

It might also be worth highlighting that current phylogenies demonstrate that parasitism is derived in metazoa and that they together with current host associations predict an origin of parasitic metazoa in the Cambrian. What you could however highlight even more is the importance a cross-section analysis brings to interpreting a negative effect and therefore proving a parasitic relationship. Analyses like yours are the way forward to demonstrate a negative in extinct organisms lacking modern analogues (De Baets and Littlewood 2015, Baumiller 2003) and not assume a negative impact if it looks like modern examples.

Whilst preservation bias and specimen abundance will always be challenging, we agree with Reviewer #1 that quantitative analyses are the way forward to accurately assess claims of potential parasitism in the fossil record. We have added a couple of sentences at the end of the opening paragraph to highlight the importance of statistical analyses in these types of intimate associations. We have also added a sentence highlighting that current phylogenies demonstrate that parasitism may have emerged in the Cambrian, as suggested by Reviewer#1.

Prevalence: One can distill the prevalence of infested versus total investigated specimens from the supplementary material, but it would worth discussing this prevalence in the main text and help to back up your interpretation of parasitism. High host specificity and/or high prevalence (between 10 up to 50-70%) are usually taken as a sign of specialist parasites, while very low (<1%) or very high prevalence (100% could signify obligate symbiosis) might signify other causes (Hengsbach 1990, De Baets, Keupp, and Klug 2015). This might also open the window for comparative studies across time and/or space and getting a better handling on collection/sampling biases. When this interaction is missing – is it because too little specimens have been sampled

(in cases encrustations are rare) or could differences have rather an evolutionary or ecological reason.

We concur with Reviewer #1 that prevalence of infestation can help to better elucidate the specific parasite-host interaction being observed. We did not however quantify the prevalence of infestation for our fauna because the specimens used for our investigation were limited to the rock slabs held in the collections of Shaanxi Key Laboratory of Early Life & Environments and Department of Geology, Northwest University. The specimens used in our analysis were not originally collected for investigations of parasite-host interactions and so do not necessarily provide an adequate sample size for accurate assessment of the frequency of infestation. The statistical methods we have used in our study to assess for potential parasitism are thus independent of infection prevalence.

The prevalence value Reviewer #1 refers to in our Supplementary data simply represents the *number of specimens* we were able to measure for our study. When compiling our dataset, we did spend significant time searching available, though limited, material for unencrusted individuals, which suggested prevalence of parasitism is greater than 50%, but more data and collections are required to more accurately constrain infestation frequency. Whilst infestation frequency data is worthy of pursuing in future research, the statistical evidence presented strongly supports our interpretation for kleptoparasitism (which is acknowledged by all reviewers).

Order of Results/Discussion: Personally, I would prefer to see first the discussion about the in-vivo encrustation and later the relationship with body size as this might make more logical sense. First you need to prove in-vivo encrustations and later the effect on hosts. If encrustations are not in-vivo they might just indicate a preference of encrusters for particular sizes of substrating. I agree that their most logical explanation is encrustation during life.

Reviewer #1 raises a relevant point here and we have reorganised the text accordingly. We demonstrate parasitism by both establishing a difference in biomass (and thus *potentially* a difference in fitness) between encrusted and non-encrusted brachiopods, by establishing that encrustation of the brachiopod host occurs in-vivo (Figure 2C) and by demonstrating that the tube apertures align with the inhalant feeding currents of the host. We follow the advice of Reviewer #1 and use both statistical evidence to establish a difference in biomass AND data establishing the in-vivo growth of the tube-dwelling organisms with the host brachiopod, to more strongly support the interpretation of kleptoparasitism in this system.

Exploration of alternative hypotheses for the size difference: you attribute the size differences almost immediately to differences in fitness but there are potentially other factors (e.g., preservational biases, selective encrustation and/or mortality of juveniles) which would be necessary to be discussed as alternative hypotheses/possibilities in main text even if you can exclude them with the data you have (Huntley and Scarponi 2012). Instead of biomass differences alone in encrusted versus non-encrusted specimens, it would be better to demonstrate an actual impact on growth through a cross-sectional analysis which your data would easily allow (Baumiller and Gahn 2018) and/or perform an actual cost-benefit analysis (Baumiller 2003).

Reviewer #1 is right that differences in biomass alone cannot be used exclusively to identify parasitic relationships in the fossil record. As noted by Reviewer #1, a range of techniques

can be used to quantify an impact on host fitness that is indicative of parasitism. Whilst impact on growth is one way to assess the presence of parasitism, it is not the only method. Our study employs multiple approaches, that do not rely solely on size differences to establish the existence of a parasite host relationship. The measurements and statistical methods used in our study are the most appropriate for the material we have at hand (see previous comment on sample size from available rock slabs) and effectively addresses the primary aims of our study. As all three reviewers have acknowledged, we have provided ample evidence to demonstrate the interaction between our tube-dwelling organism and the associated brachiopods is a parasite-host relationship.

The exceptional preservation of soft tissue and anatomical information available in our material greatly enhances the interpretation of parasitism. The preservation of delicate marginal setae and visceral tissue, for example, establishes beyond question that the brachiopods were alive with their encrusted tubes at the time of death due to smothering. The matching accretionary growth increments in host and encruster also supports an in-vivo relationship. The preservation of a spiroloph lophophore is extremely rare in the fossil record and enables the laminar currents of the inhalant stream to be reconstructed and together with the aligned orientation of the tube-dwelling organisms uniquely reinforces our interpretation of kleptoparasitism

We cannot employ either of the analyses suggested by Reviewer #1 due to the aforementioned limitations of our data. The approach utilised by Baumiller and Gahn (2018) requires an accurate value for the frequency of infestation which, for reasons previously stated, cannot be confidently attained. The cost-benefit analysis outlined in Baumiller (2003) requires some knowledge of the biological affinities of the parasite and, as we state in the main text, whilst the compressed tubes are reasonably preserved, we do not have adequate soft tissue preservation to confidently ascribe the biological affinities of the tube-dwelling parasite.

We have however considered negative impacts on growth rate in an alternate manner to that proposed in the two suggested studies. As we demonstrate, when a brachiopod is encrusted early in its life, the negative effects are greater than those brachiopods encrusted later in life (Figure 2C). This indicates there is an effect on host growth associated ***with the time of infestation***. The longer a brachiopod carries a parasite the smaller its size. Given what is known about brachiopod growth (the majority of growth occurs early in life), the impact on growth must therefore be greater if a host brachiopod is infected early in life and our results are consistent with this.

Reviewer #1 has raised the valid point that we have not been explicit in exploring alternate hypotheses for our results. We did so implicitly, but we agree with Reviewer #1 that we should directly address this issue. We now state this directly in the text.

We do not have an exact count of the number of other potential substrates we examined, but we did spend considerable time looking for tubes attached to either other substrates or free-living and were unable to find any examples of tube specimens that were not attached to brachiopods. We are confident that the tube-dwelling organisms are selective in terms of substrate choice, exclusively colonising brachiopod hosts.

These and additional suggestions can be found in the annotated pdf.

Suggested references:

- Bassett, Michael G., Leonid E. Popov, and Lars E. Holmer. 2004. "The Oldest-Known Metazoan Parasite?" *Journal of Paleontology* no. 78 (6):1214-1216.**
- Baumiller, Tomasz K, and Forest J Gahn. 2018. "The nature of the platyceratid–crinoid association as revealed by cross-sectional data from the Carboniferous of Alabama (USA)." *Swiss Journal of Palaeontology* no. 137 (2):177-187.**
- Baumiller, Tomasz K. 2003. "Evaluating the interaction between platyceratid gastropods and crinoids: a cost–benefit approach." *Palaeogeography, Palaeoclimatology, Palaeoecology* no. 201 (3–4):199-209.
doi: [http://dx.doi.org/10.1016/S0031-0182\(03\)00625-4](http://dx.doi.org/10.1016/S0031-0182(03)00625-4).**
- De Baets, K., and Littlewood, D. T. J., 2015, The Importance of Fossils in Understanding the Evolution of Parasites and Their Vectors: *Advances in parasitology*, v. 90, p. 1-51.**
- De Baets, Kenneth, Helmut Keupp, and Christian Klug. 2015. "Parasites of ammonoids." In *Ammonoid Paleobiology: From anatomy to paleoecology. Topics in Geobiology 43*, edited by Christian Klug, Dieter Korn, Kenneth De Baets, Isabelle Kruta and Royal H. Mapes, 837-875. Dordrecht: Springer.**
- Hengsbach, Rainer. 1990. "Studien zur Paläopathologie der Invertebraten. 1: Die Paläoparasitologie, eine Arbeitsrichtung der Paläobiologie." *Senckenbergiana Lethaea* no. 70 (4-6):439-461.**
- Huntley, John Warren, and Daniele Scarponi. 2012. "Evolutionary and ecological implications of trematode parasitism of modern and fossil northern Adriatic bivalves." *Paleobiology* no. 38 (1):40-51. doi: 10.1666/10051.1.**
- Peel, John S. 2014. "Failed predation, commensalism and parasitism on lower Cambrian linguliformean brachiopods." *Alcheringa: An Australasian Journal of Palaeontology*:1-15. doi: 10.1080/03115518.2015.964055.**

We thank Reviewer #1 for providing these additional references to add to our manuscript and also for the additional comments they have made on our manuscript. These additional comments have already been addressed in our previous replies above.

Reviewer #2

This is a very interesting manuscript. The authors described and discussed a unique, life association between a tube-dwelling organism and a new species of brachiopod, *Neobolus wulongqingensis* sp. nov., from the early Cambrian Guanshan Biota, Southwest China. Although symbiotic relationships have been reported several times from exceptionally well-preserved Cambrian fossils, it is generally difficult to determine the exact relationship between the symbionts if it is mutualism, commensalism or parasitism. This study took the advantage of the great abundance of available fossil specimens, cleverly applied statistical methods to explore the interaction between the brachiopod species and the associated tube-dwelling organism, and subsequently concluded this interaction as a type of parasitism (obligate kleptoparasitism). This manuscript is well prepared, the study carried out some detailed measurements and analyses, the descriptions are accurate, and both the discussions and conclusions are generally sound. Therefore, this manuscript presents a welcoming study contributing to our understanding of early symbiotic relationships from the Cambrian, and I would like to support its publication on *Nature Communications*. However, I have some

comments or questions below for further improvements.

Major points:

1) Although the authors convinced me that the interaction between the tube-dwelling organism and *N. wulongqingensis* is likely to be a parasite-host relationship, the conclusion of “obligate kleptoparasitism” might be oversold. From my understanding “obligate parasite” is a parasitic organism that cannot complete its life-cycle without exploiting a suitable host and an “obligate kleptoparasite” derives all of its energy intake from kleptoparasitism. However, we know very little about the nature and life-cycle of the tube-dwelling organisms, so there is no sufficient evidence to support “obligate kleptoparasitism”.

Reviewer #2 has highlighted an important point here. The definition of ‘obligate’ can vary dependant on context and we need to take this into account in our study. However, we do present ample evidence that the parasite cannot complete its life-cycle without a brachiopod host. As we state in the text, settlement of tube-dwelling individuals only occurs on post metamorphic brachiopod shells and is completely unknown on other substrates throughout the entire community, even though such substrates were available. This indicates high selectivity in substrate choice. We also find the growth direction of the tube-dwelling organism strongly aligns with the inhalant laminar feeding currents of the host brachiopod. Combined together, this evidence demonstrates kleptoparasitism and strongly indicates that the relationship between the parasite and its host is obligate. If alternate energy intake could sustain the tube-dwelling organism, we would not see a complete absence of individuals on alternate substrates or as free-living organisms. We also would not expect to see a complete absence of tube apertures orientated away from the brachiopod commissure.

We have now added a definition of obligate parasitism to the text to address the potential confusion highlighted by Reviewer #2 and have also removed the term ‘obligate’ from the manuscript title, given our primary focus is on demonstrating the emergence of a parasite-host animal system in an early Cambrian benthic community. We instead now use our title to highlight the novel kleptoparasitic relationship we observe.

2) Although the statistical analysis shows that the tube-dwelling organism likely reduced the fitness of infested brachiopods, it is not clear to me if this reduced fitness was entirely caused by “kleptoparasitism”. Are there any other possibilities?

For example, did the tube-dwelling organisms bore through the shell of the brachiopods?

Having examined the interior of the brachiopod valves we find no evidence of boring through the shell by the tube-dwelling organism. The parasite is encrusted to the brachiopod surface and make no direct contact with the interior of the shell. We now state this in the text.

As stated in our previous comments to Reviewer #1, it is the multiple lines of evidence we use that demonstrate kleptoparasitism. A difference in biomass alone would not be enough, but when combined with the high-fidelity preservation and the statistically significant relationship between time of attachment and the biomass of the brachiopod host along with the strongly direction orientation of the tube apertures, kleptoparasitism becomes a compelling interpretation.

3) Could the weight and attachments of these tube-dwelling organisms reduce the movement and feeding efficiency of the brachiopods? If the reduced fitness was entirely caused by being robbed food, why the increasing numbers of tube-dwelling organisms did not cause further decline of the hosts' fitness.

This is a good suggestion by Reviewer #2 and something we had also considered, but we find no evidence that the parasites inhibited feeding. If this were the case, we would expect that, as the number of parasites increases, the fitness of the host would decline. As we show in Supplementary Figures 7a-c, there is no association between increased number of parasites and host fitness. We have now updated the text to more clearly indicate that the feeding efficiency of the brachiopod is not inhibited.

4) It will be nice to introduce and compare with some other examples of interactions between epibiont and host that are observed in the living world today and in other brachiopod fossils. For example, are all these relationships parasitism? Please also see the study by Shiino & Tokuba 2016, Palaeoworld.

Unfortunately, there are limited examples of parasite-host relationships in living brachiopods. The papers indicated by Reviewer #2 focus *on post-mortem* attachment and thus are not relevant to a discussion of parasitism.

However, given this suggestion by Reviewer #2 along with suggestions by other reviewers, we have taken the opportunity to increase citations of all relevant previous papers on fossil parasitism.

5) Some parts of the manuscript lack fluidity, as different sections of the manuscript seem to be slightly disjointed. For example, the Results and Discussion went straight into the biomass analysis, which is rather abrupt from the previous section. The lack of explanation about the research background and methods can lead to many questions, such as: 1) How many specimens were analyzed? How many of them with tubes? 2) What does biomass mean here? How was it measured from fossil specimens? 3) Why need to measure biomass? Some of these questions were eventually answered in the method section, but I feel that it is still necessary to briefly explain here, or need to cite "(see Material and Methods)".

Reviewer #2 makes a series of relevant points here. We have adjusted and revised the text accordingly and following the suggestion of Reviewer #2 we have moved the Material and Methods section ahead of Results and Discussion.

Minor points:

Page 1, summary: "kleptoparasitism"

This manuscript should attract a wide audience with different background, so it will be helpful to briefly explain some terminologies when they first appear in the manuscript, such as "kleptoparasitism (parasitism by theft)".

We have made the appropriate change to the text

**Page 2, 2nd paragraph, line 5: “from other Konservat-Lagerstätten”
Change to “from other Cambrian Konservat-Lagerstätten”**

We have made the appropriate change to the text

**Page 2, 3rd paragraph, line 1: “many of the brachiopod”
It will be more accurate to explain the percentage of infested brachiopods**

We have made the appropriate change to the text

**Page 2, 3rd paragraph, line 2: “Whilst the tube-dwelling animal”
Change to “Whilst the soft body of the tube-dwelling animal”**

We have made the appropriate change to the text

**Page 2, 3rd paragraph, line 5: “(Supplementary Fig 3E-F)”
Change to “(Supplementary Fig 3E-F),”**

We have made the appropriate change to the text

**Page 2, 3rd paragraph, line 8: “indicating an intimate, life-long association”
I didn’t understand how the statement before indicate a “life-long” association? Please explain further.**

The tubes possess accretionary growth increments that parallel those of the brachiopod, indicating a close relationship between parasite and host

**Page 2, 3rd paragraph, line 9: “(Fig. 1g)”
It is better to use arrow or box to mark the trilobite fragment out in the picture**

Excluding the brachiopod host and its associated parasites, the trilobite fragment is the only other object in the image and is obviously discerned from the surrounding matrix, making an arrow or box unnecessary

**Page 3, 2nd paragraph, line 5: “tube dwelling”
Change to “tube-dwelling”, please check throughout the manuscript for consistency**

We have made the appropriate change to the text

**Page 5, 2nd paragraph, line 6: “The preferred orientation...chaetal fringe (Fig. 4)”
This sentence is too long and confusing. Please reword and divide it two sentences.**

We have made the appropriate change to the text

**Page 5, 2nd paragraph, line 10: “strongly supports kleptoparasitic behavior²³”
It is not clear why REF [23] was cited here? Please check.**

This has been corrected the text to now cite REFERENCE [42]

**Page 5, 3rd paragraph, line 10: “and the time lag...exploit this resource”
Please cite REF [24] here**

We have made the appropriate change to the text

Page 5, 3rd paragraph, line 11: “Combined with the fact the biotic...kleptoparasitic associations.”

As mentioned above, I don’t fully understand this sentence neither agree with the arguments here. Otherwise, please explain in more detail.

As we state in the text, the trophic strategy employed by brachiopods makes them particularly vulnerable to exploitation by a kleptoparasite. Combined with obligate nature of the kleptoparasite with its host, this means the effect size we observe may be larger than that for the more prevalent facultative kleptoparasitic associations. We have adjusted the text to make this make this clearer.

**Page 5, last sentence: “Intriguingly, ...during the Phanerozoic”
Same as above, this statement is over pushed**

Examples of obligate kleptoparasitism in modern marine systems are exceedingly rare. We have altered the sentence to include the possibility that obligate kleptoparasitism may have been rare in marine systems over Phanerozoic time but secondary loss some time during the Phanerozoic is also possible.

**Page 6, last few sentences: “Antagonistic biotic...Cambrian Radiation”
I can understand that the authors want to link their research to a wider implication, but these sentences are not particularly relevant to the study, and the discussion is not rigorous and convincing, more like some throw-in empty comments, so I suggest to delete these sentences; otherwise please reword.**

In our final paragraph we refer back to concepts and theory that were previously referred to in our introduction. These wider implications are directly relevant to our study, as they both provide an impetus to undertake the study in the first place and place our research in a broader context. Having now identified parasitism in a Cambrian fossil community, and given the established significance of parasite-host interactions as a driver of evolutionary change, this does raise the question as to the import of biotic interactions in driving the adaptive radiation that occurred during the Cambrian. We would be remiss in not commenting on this possibility.

Page 6, Measurements.

The whole paragraph didn’t explain what samples have been measured, how they were measured and where is the measurement data

Page 7, Imaging. “Previous studies...Dataset 2”

These sentences should be put into the above Measurements section, and some of them can be used for the Result section as well

Page 7, Statistical analysis. “Measurements of encrusted tube....in Supplementary Dataset 2.”

Move to above Measurements section

We have made the appropriate changes to the 'Materials and Methods' section to address the three previous suggestions by Reviewer #2

Reviewer #3

Dear authors,

I found that the manuscript was well constructed, and answered many of the questions that sprung to mind while reading the manuscript.

My comments and suggestions, referring to more specific and particular areas of the manuscript, are included in annotated file 'Zhang et al_Reviewer_comments', attached to my review.

You may have followed a journal template, but I thought odd the "Results and Discussion" were presented before the "Materials and Methods" section. The supplementary information file is welcome too. All images are clear and sharp; all figures/graphs are of good quality.

Regards

Yves Candela

We thank Reviewer #3 for taking the time to comment on our manuscript in detail. We have reviewed each comment and made the relevant changes to the manuscript. We have now moved the 'Materials and Methods' section ahead of the 'Results and Discussion' section as they have suggested. The majority of comments made by Reviewer #3 have been previously addressed in our reply to Reviewers #1 and #2 and appropriate changes to the text have been made.

Like Reviewer #3, we have considered discussing kleptoparasitism in the introduction of our work, but have refrained, as our study takes an agnostic approach. We do not pre-determine the symbiotic relationship we have analysed as an example of parasitism but only as a *potential* parasite-host interaction we wish to investigate. As such, it would not be consistent with this approach to explore kleptoparasitism prior to presenting our results, as there is no initial evidence to suggest the tube-dwelling organisms are kleptoparasitic. This is best presented once it is revealed by our quantitative analyses.

We also do consider the biotic relationship we observe to be novel, both because it is rare in the fossil record (but not absent, a point we have now corrected), and because it has not been previously identified in Cambrian communities. A new trophic strategy that is subsequently rare in later time periods we consider to meet the definition of novel.

Reviewers' Comments:

Reviewer #1:

Remarks to the Author:

The implemented changes make the manuscripts easier to follow. I feel you addressed all my suggestions. Looking forward to seeing this manuscript published.

Reviewer #2:

Remarks to the Author:

The authors have addressed most of my questions or suggestions during their revision or provided a sound explanation wherever they disagree. Therefore, I am happy with the current version of the manuscript and would like to support its publication.

Best wishes,
Xiaoya Ma